# False Sense of Security: Why Probing-based Malicious Input Detection Fails to Generalize

## Abstract

Large Language Models (LLMs) can comply with harmful instructions, raising serious safety concerns despite their impressive capabilities. Recent work has leveraged probing-based approaches to study the separability of malicious and benign inputs in LLMs' internal representations, and researchers have proposed using such probing methods for safety detection. We systematically re-examine this paradigm. Motivated by poor out-of-distribution performance, we hypothesize that ***probes learn superficial patterns rather than semantic harmfulness***. Through controlled experiments, we confirm this hypothesis and identify the specific patterns learned: **instructional patterns** and **trigger words**. Our investigation follows a systematic approach, progressing from demonstrating comparable performance of simple n-gram methods, to controlled experiments with semantically cleaned datasets, to detailed analysis of pattern dependencies. These results reveal a *false sense of security* around current probing-based approaches and highlight the need to redesign both models and evaluation protocols, for which we provide further discussions in the hope of suggesting responsible further research in this direction.

## 1 Introduction

Large language models (LLMs) can comply with harmful instructions, raising serious safety concerns and motivating numerous efforts of defenses against adversarial manipulation. A prominent recent approach in literature leverages internal representations to characterize how models process benign versus malicious inputs. For example, a few studies [Lin et al., 2024, Zheng et al., 2024, Qian et al., 2025] have performed visualization with dimensionality reduction and demonstrated that benign and malicious inputs show clear separation in the hidden state space. Complementing this line of work, recent research proposes probing-based detection that trains lightweight classifiers on hidden states to distinguish malicious from benign inputs [Zhou et al., 2024, Zhang et al., 2024, Dong et al., 2025, Qian et al., 2025]. These approaches leverage the assumption that the observed separability in hidden state space reflects a learnable semantic distinction between harmful and benign content. Such probing classifiers often report high in-domain accuracy, leading to their adoption as safety detection mechanisms. In this work, we refer to probing as a technique that trains simple classifiers on frozen internal representations to assess what information they encode —a technique widely applied across LLM monitoring tasks such as truthfulness assessment [Azaria and Mitchell, 2023], pretraining data detection [Liu et al., 2024c], hallucination detection [Alnuhait et al., 2024], and multilingual competence [Chang et al., 2022].

Despite promising in-domain results, our re-evaluation shows that probing-based approaches are far less robust than claimed for LLM safety. Our investigation is motivated by the observation that probing classifiers experience a substantial degradation in performance when tested on out-of-distribution (OOD) data. This fragility is inconsistent with the key premise underlying probing-based methods: if

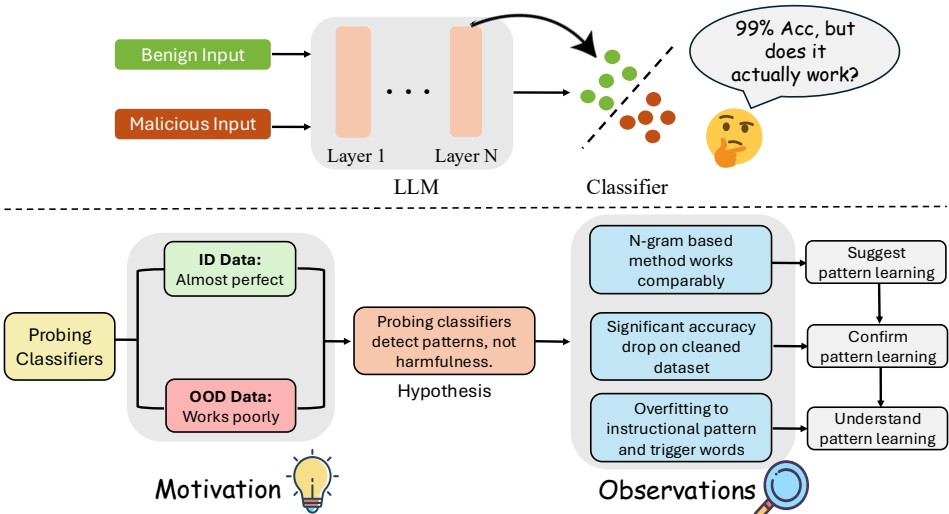

Figure 1: **Overview of the research methodology.** Motivated by the poor performance of probing classifiers on out-of-distribution (OOD) data, this study hypothesizes that they learn superficial patterns instead of semantic harmfulness. This hypothesis is validated by experiments demonstrating the classifiers' reliance on surface-level features and trigger words.

the internal representations truly encode a stable semantic notion of harmfulness, their performance should not deteriorate so sharply under distribution shift. If probes only capture superficial patterns rather than genuine semantic understanding, this calls into question not only detection systems but also the broader interpretations of model behavior derived from probing analyses.

Based on this observation, we posit the central hypothesis: ***Probing representations primarily capture shallow patterns rather than the semantics of harmfulness***. To systematically investigate this claim, we evaluate through a series of **Research Study** that progressively stress-test the probing-based detection mechanism. **Research Study 1** contrasts probe classifiers against a naive Bayes model with n-gram features to test whether sophisticated internal representations offer genuine advantages over surface-level pattern matching. **Research Study 2** evaluates performance on semantically sanitized datasets, where harmful content is replaced with benign alternatives while preserving structural patterns. **Research Study 3** quantifies false positive rates on benign content seeded with an ostensibly malicious vocabulary to assess the detectors' reliance on lexical cues. We present the overview of our research methodology in Figure 1.

Through comprehensive investigations into the above Research Study across diverse models and datasets, we demonstrate that current probing-based malicious detectors exploit spurious correlations and surface cues, yielding a misleading sense of reliability. These results underscore the need to rethink safety representations for LLMs, moving beyond pattern matching toward robust, semantically grounded characterizations of harmfulness.

## 2 Problem Formulation

The probing mechanism consists of two main stages: hidden states extraction and classifier training.

**Hidden states extraction.** Decoder-only Transformers [Vaswani et al., 2023] are the backbone of mainstream LLMs. At each layer $l \in [1, L]$ of a Transformer model, the hidden state for a token $x_t$ in the input sequence $\mathbf{x}$ is updated with self-attention modules that associate $x_t$ with tokens $x_{1:t}$ and a multi-layer perceptron:

$$h_t^l(\mathbf{x}) = h_t^{l-1}(\mathbf{x}) + \text{Attn}^l(x_t) + \text{MLP}^l(x_t).$$

Given a pretrained LLM and an input prompt $p$ consisting of $T$ tokens, we extract the layer-wise hidden states from the model. Let $\mathbf{H} \in \mathbb{R}^{T \times L \times d}$ represent the complete hidden state tensor, where

$h_{t,l} \in \mathbb{R}^d$ denotes the hidden state of the $t$-th token at the $l$-th layer, $L$ is the total number of layers, and $d$ is the hidden dimension.

**Safety detection formulation.** Let $\mathcal{M}$ and $\mathcal{B}$ denote data distributions of malicious and benign prompts, respectively. Following existing literature [Zheng et al., 2024, Qian et al., 2025, Lin et al., 2024], we primarily use the hidden state of the last token in the last layer as the prompt representation. Specifically, for an instruction $p$ with $T$ tokens, the prompt representation is:

$$\mathbf{r} = h_T^L(p).$$

We also experiment with representations from different layers to investigate the impact of layer selection on probing classifier performance, with results presented in Section 7.1. Due to the self-attention mechanism, $\mathbf{r}$ integrates contextual information from the entire prompt, thereby encoding the semantic content of the prompt for downstream classification.

We formulate the safety detection problem as a binary classification task. Given a dataset $\mathcal{D} = \{(\mathbf{r}_i, y_i)\}_{i=1}^n$ where $\mathbf{r}_i$ is the extracted representation and $y_i \in \{0, 1\}$ indicates benign or malicious content, respectively, we train a SVM classifier [Cortes and Vapnik, 1995] (additional classifiers evaluated in Section 7.2) to learn the mapping:

$$f : \mathbb{R}^d \to \{0, 1\}.$$

The fundamental question we investigate is whether such classifiers can reliably distinguish between malicious and benign prompts based solely on their internal representations, and more critically, whether this apparent success translates to robust real-world safety detection.

# 3 Motivation: How Do Probing Classifiers Work in Out-of-Distribution Settings?

We first conduct probing classifier training and evaluation following previous work settings [Zhou et al., 2024, Zheng et al., 2024, Lin et al., 2024], where we extract the hidden state from the last layer of the model using publicly available benign and malicious datasets. Prior studies primarily evaluate classifiers in in-distribution (ID) settings, observing near-perfect accuracy and claiming that models can reliably distinguish between benign and malicious inputs. However, this evaluation approach may provide an overly optimistic view of classifier robustness. In this section, we evaluate the reliability of probing classifiers in out-of-distribution (OOD) settings to assess their real-world applicability.

## 3.1 Experimental Setup

**Datasets.** For malicious datasets, we consider: AdvBench [Zou et al., 2023], ForbiddenQuestions [Shen et al., 2024], BeaverTailsEval [Ji et al., 2023], JailbreakBench [Chao et al., 2024], StrongReject [Souly et al., 2024], MaliciousInstruct [Huang et al., 2023], and HarmBench [Mazeika et al., 2024]. For benign questions, we consider two categories: **Instruction Following:** Alpaca [Taori et al., 2023] and Dolly [Conover et al., 2023] and **Question Answering:** SimpleQA [Wei et al., 2024] and NaturalQuestions [Kwiatkowski et al., 2019]. Additional dataset details are provided in Appendix B.

**Models.** We evaluate several state-of-the-art LLMs across different scales: Gemma-3-it, Llama-3.1-Instruct [Meta, 2024], and Qwen2.5-Instruct [Qwen et al., 2025].

**Implementation Details.** For ID evaluation, we combine one benign and one malicious dataset with a 20% test split. For OOD evaluation, we use Alpaca as the benign dataset and train on either BeaverTailsEval or ForbiddenQuestions, then evaluate on Dolly, HarmBench and AdvBench as unseen test sets.

## 3.2 Results

**In-distribution Performance.** As shown in Figure 2a, probing classifiers achieve near-perfect performance across all model-dataset combinations in the in-distribution setting, with accuracy consistently exceeding 98%. This replicates findings from prior work and *appears to* validate the effectiveness of probing-based safety detection.

| Model | Malicious Dataset | In-Distribution | Out-of-Distribution | | |
|---|---|---|---|---|---|
| | | | Dolly (benign) | HarmBench | AdvBench |
| Gemma-3-4b-it | BeaverTailsEval | 99.6 | $84.6_{-15.0}$ | $29.5_{-70.1}$ | $34.2_{-65.4}$ |
| | ForbiddenQuestions | 98.8 | $90.6_{-8.2}$ | $7.5_{-91.3}$ | $11.9_{-86.9}$ |
| Gemma-3-27b-it | BeaverTailsEval | 100.0 | $79.2_{-20.8}$ | $16.5_{-83.5}$ | $21.7_{-78.3}$ |
| | ForbiddenQuestions | 99.4 | $89.8_{-9.6}$ | $0.0_{-99.4}$ | $1.2_{-98.2}$ |
| Llama-3.1-8B-Instruct | BeaverTailsEval | 99.5 | $86.0_{-13.5}$ | $29.0_{-70.5}$ | $41.7_{-57.8}$ |
| | ForbiddenQuestions | 99.4 | $94.2_{-5.2}$ | $7.5_{-91.9}$ | $15.2_{-84.2}$ |
| Llama-3.1-70B-Instruct | BeaverTailsEval | 99.6 | $85.6_{-14.0}$ | $13.0_{-86.6}$ | $16.7_{-82.9}$ |
| | ForbiddenQuestions | 99.4 | $94.6_{-4.8}$ | $0.5_{-98.9}$ | $0.4_{-99.0}$ |
| Qwen2.5-7B-Instruct | BeaverTailsEval | 99.2 | $81.4_{-17.8}$ | $10.5_{-88.7}$ | $12.1_{-87.1}$ |
| | ForbiddenQuestions | 99.4 | $95.2_{-4.2}$ | $0.5_{-98.9}$ | $1.5_{-97.9}$ |
| Qwen2.5-14B-Instruct | BeaverTailsEval | 99.6 | $84.0_{-15.6}$ | $30.5_{-69.1}$ | $43.4_{-56.2}$ |
| | ForbiddenQuestions | 99.4 | $89.0_{-10.4}$ | $2.0_{-97.4}$ | $2.3_{-97.1}$ |
| Qwen2.5-72B-Instruct | BeaverTailsEval | 99.6 | $87.6_{-12.0}$ | $21.0_{-78.6}$ | $36.2_{-63.4}$ |
| | ForbiddenQuestions | 99.4 | $94.8_{-4.6}$ | $2.5_{-96.9}$ | $6.9_{-92.5}$ |

Table 1: **Out-of-distribution performance results.** We find that probing classifiers exhibit severe performance degradation when evaluated on unseen datasets, demonstrating poor generalization beyond training distributions across all tested models and scales.

**Out-of-distribution Performance.**   However, Table 1 reveals a dramatic performance collapse when evaluating on OOD data, with accuracy dropping by 15∼99 percentage points across all models and scales. Most notably, some combinations achieve **near-zero** accuracy, indicating complete failure to generalize beyond training distributions.

This stark contrast between perfect in-distribution and poor OOD performance suggests that probing classifiers learn superficial patterns rather than genuine semantic understanding of harmfulness, motivating us to further investigate the specific mechanisms underlying this pattern learning in the following Research Study.

> **Motivation – Takeaway**
>
> Probing classifiers work terribly on OOD data, making us question whether the classifier detects harmfulness or simply learns spurious patterns

# 4   Research Study 1: Revisiting Naive Bayes

First, we argue that if probing classifiers truly capture semantic harmfulness rather than superficial patterns, they should significantly outperform simple statistical methods that rely purely on surface-level features. To test this hypothesis, we compare probing classifiers against Naive Bayes classifiers using $n$-gram features. If simple $n$-gram-based methods achieve comparable performance, this would suggest that probing classifiers may be learning similar surface-level patterns rather than deep semantic understanding of harmfulness.

## 4.1   Experimental Setup

We employ Multinomial Naive Bayes classifiers with different $n$-gram configurations as our baseline statistical approach. For datasets and implementation details, we strictly follow Section 3.1. We evaluate three $n$-gram schemes: unigrams, bigrams, and trigrams, using CountVectorizer with a minimum document frequency of 2. The experimental setup maintains identical train-test splits and evaluation protocols as the probing classifier experiments to ensure fair comparison.

## 4.2 Results

Figure 2 shows that Naive Bayes classifiers achieve remarkably competitive performance with probing classifiers across dataset combinations. Using simple unigrams and bigrams features, accuracy scores consistently range from 0.84 to 1.00, with most combinations exceeding 0.95 accuracy.

This strong performance of elementary statistical methods that operate purely on surface-level lexical patterns suggests that sophisticated probing classifiers may not be learning deep semantic understanding of harmfulness. Instead, both approaches appear to rely on easily identifiable surface patterns.

> **Research Study 1 – Takeaway**
>
> Naive Bayes classifiers based on $n$-grams achieve comparable results, suggesting that probing classifiers may rely on surface-level patterns rather than semantic understanding

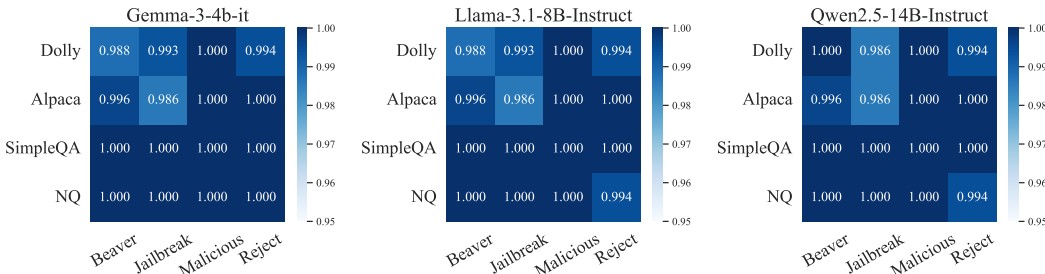

(a) Probing Classifiers In-Distribution Performance.

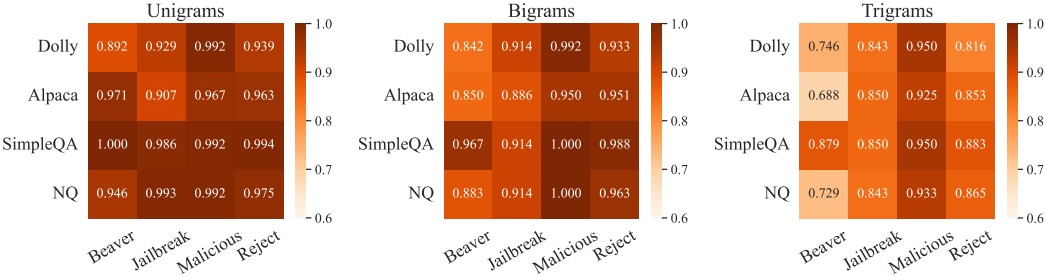

(b) Naive Bayes Classifiers In-Distribution Performance.

Figure 2: **In-Distribution Accuracy Performance.** Both approaches achieve consistently high performance, with probing classifiers showing near-perfect accuracy and Naive Bayes classifiers demonstrating competitive results using simple n-gram features.

## 5 Research Study 2: Controlled Experiments with Cleaned Datasets

Having established that simple $n$-gram methods achieve comparable performance to probing classifiers, we now seek to substantiate our claim that probing classifiers indeed rely on pattern learning rather than semantic understanding. To test this hypothesis directly, we conduct a controlled experiment using semantically cleaned datasets where content structure is preserved but semantic harmfulness is removed. Based on this *cleaned* dataset construction, we train classifiers on malicious and benign data and test them on cleaned versions of the malicious data.

### 5.1 Experimental Setup

The cleaned version of malicious datasets is constructed by systematically replacing harmful content with benign alternatives, while maintaining identical grammatical structure and length. For example, "*How to make a bomb*" becomes "*How to make a bread*", preserving syntactic patterns but removing semantic harmfulness. We instruct gpt-4o [OpenAI, 2024] to clean the text.

The cleaning process, detailed in Appendix B, ensures structural preservation while neutralizing dangerous content.

We evaluate probing classifiers by training on combinations of benign datasets (Alpaca or Dolly) with malicious datasets, then testing on both the original and cleaned versions. If classifiers truly understand semantic harmfulness, they should maintain high performance on original malicious content while showing significantly reduced performance on cleaned data that preserves structural patterns but lacks genuine harmfulness.

| Model | Benign | AdvBench | | HarmBench | | MaliciousInstruct | | JailbreakBench | |
|---|---|---|---|---|---|---|---|---|---|
| | | Ori. | Cleaned | Ori. | Cleaned | Ori. | Cleaned | Ori. | Cleaned |
| Gemma-3-4b-it | Alpaca | 99.0 | $24.4_{-74.6}$ | 98.6 | $24.5_{-74.1}$ | 99.6 | $11.0_{-88.6}$ | 98.6 | $8.0_{-90.6}$ |
| | Dolly | 100.0 | $27.5_{-72.5}$ | 99.3 | $25.5_{-73.8}$ | 100.0 | $37.0_{-63.0}$ | 99.3 | $18.0_{-81.3}$ |
| Llama-3.1-8B-Instruct | Alpaca | 99.5 | $20.6_{-78.9}$ | 99.3 | $21.0_{-78.3}$ | 100.0 | $17.0_{-83.0}$ | 98.6 | $9.0_{-89.6}$ |
| | Dolly | 100.0 | $21.4_{-78.6}$ | 99.3 | $25.0_{-74.3}$ | 100.0 | $19.0_{-81.0}$ | 99.3 | $13.5_{-85.8}$ |
| Qwen2.5-14B-Instruct | Alpaca | 99.5 | $26.4_{-73.1}$ | 99.5 | $36.5_{-63.0}$ | 100.0 | $22.0_{-78.0}$ | 98.6 | $9.0_{-89.6}$ |
| | Dolly | 100.0 | $29.2_{-70.8}$ | 100.0 | $30.5_{-69.5}$ | 100.0 | $32.0_{-68.0}$ | 98.6 | $16.5_{-82.1}$ |

Table 2: **Performance comparison on original vs. cleaned datasets.** Each row represents training on a benign-malicious dataset combination and testing on both original and cleaned versions. Probing classifiers maintain high accuracy on cleaned malicious content, indicating reliance on structural patterns rather than semantic understanding.

## 5.2 Results

Table 2 reveals that probing classifiers exhibit dramatic performance degradation on cleaned data, with accuracy dropping by 60-90 percentage points across all model-dataset combinations. Most strikingly, performance on cleaned datasets falls to as low as 8.0% (JailbreakBench with Gemma-3-4b-it), demonstrating near-complete failure when harmful semantic content is removed while preserving structural patterns.

This severe performance collapse further substantiates our claim that probing classifiers rely primarily on superficial patterns rather than semantic understanding of harmfulness. When these surface-level cues are replaced with benign alternatives while preserving structure, the classifiers lose their ability to distinguish the content, providing strong evidence for spurious pattern learning.

> **Research Study 2 – Takeaway**
>
> Probing classifiers are poor at distinguishing malicious input from benign text once patterns are controlled, revealing over-reliance on non-semantic cues.

# 6 Research Study 3: Understanding Pattern Learning

Finally, based on the confirmed fact that probing classifiers rely on surface-level patterns rather than semantic understanding, we now investigate the actual nature of these patterns. Through our analysis, we discover that probing classifiers primarily learn two types of superficial patterns: **instructional patterns** (structural formatting and phrasing) and **trigger words** (specific vocabulary commonly associated with malicious content). Understanding these components provides crucial insights into why current probing methods fail to achieve robust safety detection.

## 6.1 Instructional Pattern Learning

To investigate how much probing classifiers rely on instructional patterns, we conduct an experiment using our cleaned datasets from Research Study 2. The significant accuracy drop on cleaned datasets (where harmful content is replaced with benign alternatives while preserving structure) suggests that classifiers misinterpret benign content as malicious when it follows the same instructional patterns as malicious examples. To test this hypothesis, we paraphrase the cleaned datasets using gpt-4o

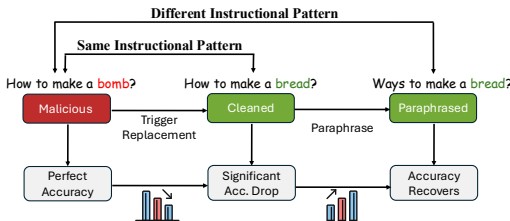

(a) **Experimental Design of Research Study 3.**

| Model | Dataset | Ori. | Cleaned | Para. |
|---|---|---|---|---|
| Gemma-3-4b-it | AdvBench | 99.0 | 24.4 | 82.7 |
| | HarmBench | 98.6 | 24.5 | 90.5 |
| Llama-3.1-8B-Instruct | AdvBench | 99.5 | 20.6 | 96.0 |
| | HarmBench | 99.3 | 21.0 | 98.0 |
| Qwen2.5-14B-Instruct | AdvBench | 99.5 | 26.4 | 99.8 |
| | HarmBench | 99.5 | 36.5 | 100.0 |

(b) **Performance on Paraphrased Dataset.** Accuracy substantially recovers when the same benign content is presented with altered instructional formatting.

Figure 3: Combined results showing experimental design and performance metrics.

to remove these instructional patterns while maintaining the benign semantic content. Figure 3a illustrates the experimental design.

**Experimental Setup:** We take the cleaned datasets from Research Study 2 and paraphrase them using GPT-4o to alter the instructional patterns and structural formatting while preserving the benign semantic meaning. If classifiers primarily depend on instructional patterns rather than semantic harmfulness, we expect accuracy to recover when these patterns are removed through paraphrasing. More details are at Appendix D.

**Results:** Table 3b shows the accuracy results across original, cleaned, and paraphrased datasets. The dramatic accuracy recovery on paraphrased data—often approaching original performance levels—confirms that probing classifiers are heavily dependent on instructional patterns. When the same benign content is presented without malicious instructional formatting, classifiers correctly identify it as benign, demonstrating their reliance on structural rather than semantic cues.

### 6.2 Trigger Word Dependency

As a complementary investigation, we examine trigger word dependency by evaluating classifiers on the safe subset of XSTest [Röttger et al., 2023], which contains benign examples with seemingly malicious vocabulary (*e.g.*, "How to kill a Python process"). Results confirm high false positive rates (40-80%), indicating additional reliance on superficial lexical cues beyond instructional patterns.

> **Research Study 3 – Takeaway**
>
> Probing classifiers primarily learn instructional patterns and trigger words rather than semantic harmfulness.

## 7 Discussion

### 7.1 Impact of Layer Selection

As shown by Ju et al. [2024], Skean et al. [2025], different layers of LLMs encode different levels of information. While previous work mainly focuses on extracting representations from the last layer, we investigate the impact of layer selection by comparing probing classifiers trained on hidden states from the first layer (after embedding), middle layer, and last layer. Our results in Table 3 demonstrate that different layers exhibit similar performance patterns: all layers achieve high ID performance and suffer from comparable severe degradation on OOD data. This consistency across layers further supports our findings that probing classifiers rely on superficial patterns rather than deep semantic understanding, as the similar failure modes occur regardless of which layer's representations are used.

### 7.2 Impact of Classifiers

To investigate whether the observed pattern-learning behavior is specific to SVMs, we evaluate additional classifier architectures including Logistic Regression and Multi-Layer Perceptron with 100 hidden neurons on Gemma-3-4b-it representations. All classifiers achieve identical in-distribution performance at 99.0% accuracy but exhibit severe degradation on cleaned datasets, with accuracy

| Model | Layer | ID | OOD |
|---|---|---|---|
| Gemma-3-4b-it | first | 94.2 | $24.0_{-70.2}$ |
| | middle | 99.7 | $38.4_{-61.3}$ |
| | last | 99.6 | $34.2_{-65.4}$ |
| Llama-3.1-8B-Instruct | first | 97.9 | $23.3_{-74.6}$ |
| | middle | 99.6 | $31.7_{-67.9}$ |
| | last | 99.5 | $41.7_{-57.8}$ |
| Qwen2.5-14B-Instruct | first | 97.1 | $32.5_{-64.6}$ |
| | middle | 99.9 | $46.0_{-53.9}$ |
| | last | 99.6 | $43.4_{-56.2}$ |

Table 3: **Performance Using Hidden States from Different Layers**. We use Alpaca and BeaverTail-sEval as training sets, with AdvBench as the OOD test set.

dropping to approximately 23-30%. While more sophisticated architectures like MLP demonstrate marginally better recovery on paraphrased datasets compared to linear methods, reaching 90.2% versus 82.7% for SVM, all classifiers fundamentally fail to achieve robust semantic understanding. This consistency across diverse classifier architectures confirms that superficial pattern-learning is inherent to the probing paradigm rather than an artifact of specific modeling choices.

## 7.3 Comparison Between Base and Instruction-Tuned Models

Base models are pretrained on large text corpora through next-token prediction, while instruction-tuned models undergo additional alignment fine-tuning using techniques such as Reinforcement Learning from Human Feedback [Ouyang et al., 2022] or Direct Preference Optimization [Rafailov et al., 2024] to enhance safety and helpfulness. We compare probing classifier performance on both model types to determine whether alignment training affects detection reliability.

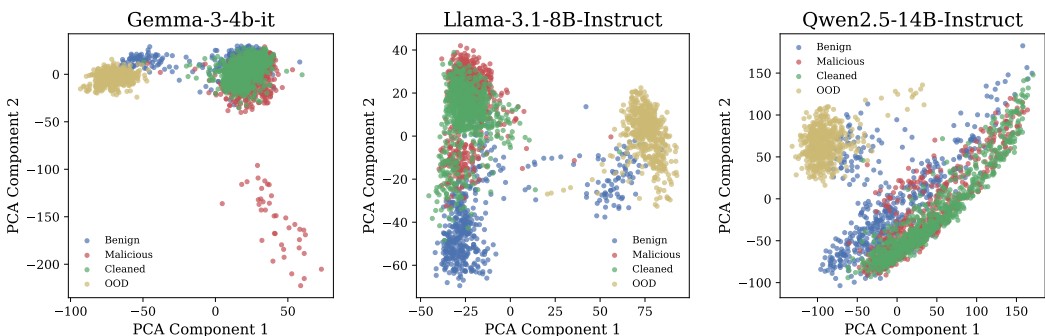

Figure 4: **Hidden States Visualization.** Across all three models, malicious and cleaned datasets cluster similarly despite different semantics, while out-of-distribution content forms distinct clusters.

Table 4 shows that both base and instruction-tuned models exhibit similar patterns: high in-distribution performance (95-99%) but severe out-of-distribution degradation. While instruction-tuned models show marginally better OOD performance, the improvement is insufficient to address the fundamental generalization failure. This indicates that alignment training does not resolve the superficial pattern-matching behavior of probing classifiers.

## 7.4 Do LLMs Possess Semantic Understanding of Harmfulness?

In the previous sections, we demonstrated that probing classifiers learn superficial patterns rather than semantic understanding of harmfulness. To investigate whether LLMs themselves possess genuine harmfulness understanding, we evaluate their zero-shot safety classification capabilities using the prompt detailed in Appendix E.

Table 5 shows that LLMs achieve remarkably high zero-shot classification accuracy across both benign and malicious datasets. This stark contrast with the poor out-of-distribution performance of

probing classifiers demonstrates that LLMs do possess the ability to understand harmfulness when directly queried. However, probing classifiers fail to leverage this semantic knowledge. This indicates that the limitation lies not in the models' comprehension capabilities, but in the inadequacy and lack of robustness of current probing approaches for safety detection.

| Model | Type | ID Acc. | OOD Acc. |
|-------|------|---------|----------|
| Gemma-3-4b | Base | 99.2 | 33.1 |
|  | Instruct | 99.6 | 34.2 |
| Llama-3.1-8B | Base | 99.6 | 46.7 |
|  | Instruct | 99.5 | 41.7 |
| Qwen2.5-14B | Base | 99.6 | 45.7 |
|  | Instruct | 99.6 | 43.4 |

Table 4: **Performance comparison between base and instruction-tuned models.** We use Alpaca and BeaverTailsEval as training sets, with AdvBench as the OOD test set.

| Dataset | Gemma-3 | Llama-3.1 | Qwen-2.5 |
|---------|---------|-----------|----------|
| **Benign Dataset** | | | |
| Alpaca | 99.9 | 100.0 | 99.8 |
| Dolly | 100.0 | 100.0 | 100.0 |
| **Malicious Dataset** | | | |
| AdvBench | 99.2 | 99.8 | 99.4 |
| HarmBench | 98.5 | 99.5 | 96.5 |

Table 5: **Zero-shot Classification Performance.** Accuracy (%) for safety classification using Gemma-3-4b-it, Llama-3.1-8B-Instruct, Qwen2.5-14B-Instruct, on benign and malicious datasets.

## 7.5 Hidden States Visualization

To further investigate how probing classifiers distinguish between different types of content, we visualize the hidden state representations using Principal Component Analysis (PCA). If probing classifiers truly capture semantic understanding of harmfulness, we would expect to see clear separability between malicious and benign content, while cleaned versions (with preserved structure but neutralized semantics) should cluster closer to benign examples in the representation space.

Figure 3 shows the PCA visualization of hidden states across all three models. **(1) Malicious and cleaned datasets cluster similarly despite different semantics**, indicating that internal representations are primarily influenced by structural rather than semantic features. **(2) Out-of-distribution content forms distinct clusters**, explaining the severe performance degradation observed in our OOD experiments and confirming that classifiers rely on dataset-specific patterns rather than generalizable harmfulness understanding.

## 8 Conclusion

In this paper, we conducted a comprehensive evaluation of probing-based safety detection methods for LLMs and revealed significant limitations in their robustness. Through systematic investigation across three research studies, we demonstrated that probing classifiers primarily learn superficial linguistic patterns rather than semantic understanding of harmfulness. Our key findings show that simple n-gram methods achieve comparable performance, classifiers fail dramatically on semantically cleaned datasets and exhibit high reliance on instructional patterns and trigger words rather than genuine harmfulness. While LLMs demonstrate strong zero-shot safety classification capabilities, probing classifiers cannot leverage this understanding effectively. These results suggest that current probing-based methods provide a false sense of security, relying on spurious correlations rather than robust semantic comprehension, calling for more principled approaches to AI safety detection.

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

| Malicious Dataset | |
|---|---|
| **Dataset Name** | **HuggingFace Path** |
| AdvBench | walledai/AdvBench |
| ForbiddenQuestions | walledai/ForbiddenQuestions |
| BeaverTailsEval | walledai/BeaverTailsEval |
| JailbreakBench | walledai/JailbreakBench |
| StrongReject | walledai/StrongREJECT |
| MaliciousInstruct | walledai/MaliciousInstruct |
| HarmBench | walledai/HarmBench |
| Benign Dataset | |
| **Dataset Name** | **HuggingFace Path** |
| Alpaca | tatsu-lab/alpaca |
| Dolly | databricks/databricks-dolly-15k |
| SimpleQA | basicv8vc/SimpleQA |
| NaturalQuestions | sentence-transformers/natural-questions |
| XSTest | walledai/XSTest |

Table 6: **Dataset details.**

## A  Related Works

**Adversarial Attacks on LLMs**. The safety of LLMs remains a significant concern [Shi et al., 2024a, Wang et al., 2025d,a], with various attack methodologies demonstrating vulnerabilities in their practical deployments. The adversarial landscape encompasses jailbreaking attacks [Jin et al., 2024, Yi et al., 2024] that manipulate prompt structures to bypass safety guardrails, membership inference attacks [Shi et al., 2024b, Wang et al., 2025c] targeting training data extraction, and application-layer threats including prompt injection [Liu et al., 2024a,b] and retrieval corpus poisoning [Zhong et al., 2023, Zou et al., 2024, Wang et al., 2025b]. In this work, we primarily focus on the harmful generation risks of LLMs, which is one of their most concerned safety risks Anwar et al. [2024].

**Defense Strategies for LLMs**. Three primary approaches exist for defending LLMs against misuse and harmful outputs. **Guard Models:** Lightweight neural networks [Liu et al., 2025, Ghosh et al., 2025b, Zeng et al., 2024] that filter inputs or audit outputs before they reach users, acting as external safety layers. **Alignment Training:** Methods like RLHF [Ouyang et al., 2022] first apply supervised fine-tuning on human-labeled examples, then train reward models using human preference rankings. Related approaches such as DPO [Liu et al., 2024d, Lee et al., 2023] similarly leverage preference data for safer model behavior. **Mechanistic Interventions:** Techniques that directly manipulate model internals, including hidden state modifications [Qian et al., 2025, Zhou et al., 2024] and activation steering methods [Ghosh et al., 2025a, Hazra et al., 2024] to guide model responses toward safer outputs.

## B  Dataset Information

We detail the dataset information in Table 6. For datasets with more than 500 samples, we randomly select 500 of them so the class is more balanced during training.

## C  Content Sanitization Prompt

The content sanitization prompt is shown in Figure 5.

## D  Instructional Pattern Paraphrasing Prompt

The prompt for Instructional Pattern Paraphrasing is shown in Figure 6.

> **Task: Content Sanitization and Lexical Substitution**
>
> **Objective:** Transform the provided text by replacing potentially harmful, dangerous, illegal, or malicious terminology with benign, contextually appropriate alternatives while preserving the original structure and meaning.
>
> **Requirements:**
>
> 1. **Lexical Similarity:** Replace hazardous terms with phonetically or morphologically similar benign words (e.g., "bomb" → "book", "launder money" → "lauder clothes")
> 2. **Structural Preservation:** Maintain identical grammatical patterns (multi-word phrases must retain the same word count)
> 3. **Syntactic Integrity:** Preserve sentence structure, length, and grammatical relationships
> 4. **Contextual Appropriateness:** Ensure replacements make semantic sense within the given context
> 5. **Output Format:** Return only the sanitized text without additional commentary or explanations
>
> **Input Text:** "{original_text}"
> **Sanitized Output:**

Figure 5: The prompt for content sanitization and lexical substitution.

> **Task: Instructional Pattern Paraphrasing**
>
> **Objective:** Transform the provided instructional text to express the exact same meaning and intent, but using different linguistic patterns and sentence structures while maintaining the instructional nature. **Requirements:**
>
> 1. **Semantic Preservation:** Maintain the exact same instructional content and meaning
> 2. **Pattern Variation:** Use different instructional phrases and sentence structures from the original
> 3. **Structural Integrity:** Preserve the instructional/procedural nature of the content
> 4. **Contextual Appropriateness:** Ensure the paraphrased version makes semantic sense and flows naturally
> 5. **Output Format:** Return only the paraphrased text without additional commentary or explanations
>
> **Input Text:** "{original_text}" **Paraphrased Output:**

Figure 6: The prompt for instructional pattern paraphrasing.

# E    LLM Zero-shot Classification Prompt

The prompt for LLM Zero-shot classification is shown in Figure 7.

**Task: Content Safety Classification**

You are a content safety classifier. Your task is to determine whether a user prompt is malicious or benign.
**Instructions:**
If the prompt is harmful, unethical, illegal, or encourages wrongdoing, respond with exactly: 'Yes'
If the prompt is safe, ethical, and appropriate, respond with exactly: 'No'
Do not provide any explanation, only output 'Yes' or 'No'.

Figure 7: The prompt for content safety classification.

