# OpenReview forum: "False Sense of Security: Why Probing-based Malicious Input Detection Fails to Generalize"
_NeurIPS.cc/2025/Workshop/Reliable_ML — NeurIPS 2025 - Reliable ML Workshop_

### Official Review · Reviewer_4SDx · 2025-09-14
**Interesting experiments, moderately interesting conclusions**

**Rating:** 5
**Confidence:** 3

**Review:**

-- Summary. This paper provides experimental evidence for the claim that probing classifiers do not necessarily identify semantic harmfulness, but, they merely identify syntactic patterns and trigger words. In particular, the authors provide three experimental setups:
1. They compare the in-distribution performance of simplistic models with the in-distribution performance of probing classifiers, and find that these are similar.
2. They compare the OOD performance of probing classifiers on benign prompts that are, however, designed to be similar in structure to malicious ones, and find that these are very often wrongly classified as harmful.
3. They change the syntactic structure of these benign prompts and find that the harmful prompt detection accuracy increases, indicating that probing classifiers are sensitive to the syntactic structure of the prompts.

-- Strengths. Understanding how to identify harmful prompts is a crucial problem for reliable and safe use of LLMs.

-- Weaknesses. While the premise of this paper is interesting, I believe that the findings not very compelling for the following reasons.
1. While the title refers to a false sense of security, the experiments only identify scenarios where the probing classifiers make false positive predictions. I would term this as "false sense of usefulness", rather than security, because the authors have not demonstrated cases where the probing classifiers systematically accept prompts that are harmful.
2. The interpretation of Research study 1 does not seem very convincing. In particular, the fact that simple n-gram models achieve good in-distribution performance does not necessarily indicate that probing classifiers are not useful. It seems plausible, for example, that it just so happens that the harmful prompts in the training distribution are significantly different from benign prompts in terms of syntax and vocabulary, making the problem easy even for simple models. In this scenario, the method of probing classifiers could still be better than n-grams, if the training dataset were to be curated appropriately. A more convincing comparison would be between the OOD performance of n-grams and of probing classifiers.
3. The authors do not suggest a path towards harmful prompt detection. For example, inspired by the previous point regarding research study 1, one interesting question might be whether using a simple n-gram model to remove harmful prompts that can be detected easily and train a probing classifier on the remaining examples would yield improved OOD detection.

Additional comment: In lines 151-154 and the caption of Table 2, it seems that the authors state that the classifier should perform poorly on the clean dataset if it were actually detecting harmful content, which seems false. In particular, the classifier does perform poorly on the clean dataset according to the results in Table 2, which actually indicates that it does not really detect harmful content, rather than the other way around. Is this a typo?

---

### Official Review · Reviewer_KgqU · 2025-09-17
**Good paper**

**Rating:** 8
**Confidence:** 4

**Review:**

1. Summary: The paper hypothesizes that probe-based methods of language model representations to detect malicious input learn superficial patterns rather than actual malicious semantics. The authors proceed to demonstrate the hypothesis step-by-step, by showing that probe-based classifiers have poor out-of-distribution (OOD) performance; have a similar performance with n-gram-based naive bayes models, suggesting that they rely on surface-level patterns; are poor at classifying once patterns are controlled; and primarily learn patterns and trigger words.

2. Strengths: Although probe-based methods are not an entirely novel topic, as the authors themselves cite some recent works on this topic, the authors demonstrate through a series of well-designed and controlled experiments that the efficacy of such methods can be overstated. The key strengths of the paper lie in the straightforward research question(s) and their motivations, as well as the well-designed experiments.

3. Weaknesses: Some minor rewrites might be needed. For example, figure 3b should mention that alpaca was used as the benign dataaet somewhere. In table 3, what does middle layer mean? Is it layer 16 in a 32 layer transformer? Or is it a combination of middle layer(s)?